# Marine Boundary Layer Height Obtained by New Numerical Regularization Method Based on GPS Radio Occultation Data

**DOI:** 10.3390/s20174762

**Published:** 2020-08-23

**Authors:** Jianyin Zhou, Jie Xiang, Sixun Huang

**Affiliations:** 1College of Meteorology and Oceanography, National University of Defense Technology, Nanjing 211101, China; zhoujianyin17@nudt.edu.cn (J.Z.); huangsixun2021@163.com (S.H.); 2State Key Laboratory of Satellite Ocean Environment Dynamics, Second Institute of Oceanography, State Oceanic Administration, Hangzhou 361005, China

**Keywords:** marine boundary layer height, numerical regulation method, double-parameter model function method, COSMIC data, climate characteristics

## Abstract

The boundary layer height (BLH) determines the interface between the lower and the free atmosphere, and it is a key variable in numerical simulations and aerosol and environmental pollution studies. This article proposes a novel method in conjunction with numerical regularization to analyze the climate characteristics of the marine boundary layer height (MBLH) using 2007–2011 GPS-RO data from the COSMIC mission. The MBLH corresponds to the smallest gradient, which is calculated using the numerical regulation method where the regulation parameters are determined by the double-parameter model function method. The results reveal the relationship between the MBLH and ocean currents for the first time. A low MBLH is associated with cold seasons and seas where cold ocean currents prevail whereas a high MBLH is related to warm seasons and seas where warm currents prevail. This correlation was validated by comparing the obtained results with different occultation data including atmprf and echprf, which also showed that atmprf is more sensitive to convective cloud top capture. To test the credibility of the results, the standard deviation was used to express the MBLH confidence level. The results show that the standard deviation of the MBLH was highest in low latitudes and lowest in the middle and high latitudes. Furthermore, we analyzed the trends in interannual MBLH variations, which display significant seasonal variations and spatial distributions that correspond with the current and subsolar point. Finally, we conducted a case study in the South China Sea, and identified a distinctive seasonal change and interannual decline in MBLH.

## 1. Introduction

The physical processes that take place in the marine atmospheric boundary layer mostly occur at the sub-grid scale [1,2], where sensible heat flux, water vapor flux, turbulent vertical transport, sea breeze intensity, and sea waves impact the marine boundary layer height (MBLH) [3]. Thus, accurately analyzing the MBLH climate characteristics is particularly important, as MBLH is a key variable for the boundary layer parameterization scheme in the climate numerical model.

In many recent studies, the data used to detect the boundary layer height include radiosonde data [4,5,6], various types of radar data [7,8,9,10,11], ERA-Interim reanalysis data [12], and GPS occultation data, etc. Different data have their own merits and drawbacks. Radar and radiosonde data are more accurate and have high detection frequency in the fixed area, but are inappropriate for climate change analysis at the MBLH due to their limited spatial distribution. Also, ERA-Interim data are model-based reanalysis data, which assimilate background field data, radiosonde data, etc., but have coarse resolution and inherent model errors. However, data from the Formosa Satellite Mission 3 (FORMOSAT-3)/Constellation Observing System for Meteorology, Ionosphere, and Climate (COSMIC) satellites have a resolution of ~100 m at 0 m–1 km, ~200 m at 1–2 km, and ~300–400 m at 2–5 km. Although the accuracy of the data in the lower atmosphere is easily affected by weather and topography, its merits include high vertical resolution, and all-weather and global exploration, which compensates for the lack of observatory data in the ocean area.

GPS radio occultation observation technology utilizes the radio waves’ additional phase delay and amplitude change in the atmosphere, which is caused by the interaction of electromagnetic waves emitted from high and low orbit satellites. Furthermore, it can also obtain the bending angle profile of the radio waves due to the atmospheric refraction gradient. Assuming local spherical symmetry of the Earth’s atmosphere, we can know that the bending angle and refractivity have a one-to-one mapping relationship (Equation (1)), and the nonlocal refractive index can be obtained via Equation (2). In the process of obtaining the bending angle and refractivity, we found that the former was more advantageous than the latter when calculating the ABL height. First, the bending angle data is obtained directly from the original, observed optical path length, which has relatively limited observational error characteristics and is more sensitive to the vertical distribution of meteorological elements in the atmosphere. Second, it offers an alternative to calculating the refractivity profile using the Abelian inverse integral. Thus, it avoids diffusion of the bending angle observation error on the refractivity, which is a side effect of a poorly posed Abelian integral, and circumvents the unnecessary error caused by the Abelian weak singular kernel in the process of numerical discrete calculation. So, the COSMIC occultation bending angle data can be reliably used to characterize global MBLH climate change.
(1)$$\alpha (a)=-2a\int_{r_{0}}^{\infty }\frac{\frac{d\ln n}{dr}}{\sqrt{n^{2}r^{2}-a^{2}}}dr$$
(2)$$n^{\mathrm{GPS}}(r)=\exp\left(\frac{1}{\pi }\int_{r}^{\infty }\frac{\alpha (x)}{\sqrt{x^{2}-a^{2}}}dx\right)$$

Given that the boundary layer height (i.e., the top of the boundary layer) plays a role in improving weather forecast accuracy, climate prediction, and air quality research [5,11,13,14], numerous studies have recently been conducted on boundary layer height changes. Basha and Ratnam [15] used the maximum gradient method in conjunction with 2.5 years of sounding data from a tropical station at Gadanki to determine the boundary layer height and to study the seasonal and daily changes at the station. Ao, et al. [16] used the COSMIC/FORMOSAT-3 occultation data from 2006–2009 and the European Centre for Medium-Range Weather Forecasts (ECMWF) data (ERA-Int) to determine the boundary layer height by assessing the minimum refractivity height and water vapor pressure gradient, and then used its significance coefficient, as proposed in 2008, as the basis to test the credibility of the results. Yehui, et al. [6] used the bulk Richardson number method to determine the boundary layer height, then they made a rough estimate of the boundary layer height change trend throughout the European region. Chan and Wood [17] improved on the maximum gradient method proposed by Sokolovskiy, et al. [18], and analyzed the seasonal cycle characteristics of the global boundary layer height; Shu-peng, et al. [19] analyzed the seasonal and daily changes in the MBLH of the Southeast Pacific Ocean; Guo, et al. [5] used the bulk Richardson number method in conjunction with 1976–2016 radiosonde data from China Radiosonde Network (CRN) to determine the boundary layer height, then studied the boundary layer height spatiotemporal variation trend over China. Chien, et al. [20] studied the MBLH over the Western North Pacific (WNP) based on COSMIC profiles in addition to three other sources of data. In this work, we proposed a novel method—a numerical differential model function method for determining the boundary layer height, then we applied it to examine the MBLH global seasonal variation characteristics and interannual changes near the South China Sea.

Unlike the BLH on land, which exhibits significant diurnal fluctuations, the MBLH changes relatively slowly over space and time because the ocean serves as its underlying surface. Essentially, the strong mixing of seawater makes the physical properties of the surface uniform. In addition, due to its massive heat capacity, even if a large amount of heat from the sun is absorbed, the temperature of the underlying surface, which forces the boundary layer to rise, does not change significantly [3]. Drastic changes in the MBLH mostly occur in small and medium scale weather systems in response to different air masses on the sea surface undergoing vertical movement and convection under forcing. Such processes cause increased turbulent mixing in the mixed layer, thereby raising the MBLH [21].

In this work, we used the numerical differential regularization method combined with COSMIC occultation bending angle data to analyze the trends in MBLH due to climate change in the global ocean. We begin by introducing the data and methods used in our research, then we demonstrate the global distribution of MBLHs, their seasonal variation, and interannual variation characteristics and finally, we summarize our findings.

## 2. Data

In this study, we employed GPS radio occultation (GPS-RO) data from the COSMIC mission. The data was obtained from the COSMIC RO Data Analysis and Archive Center (CDAAC: https://cdaac-www.cosmic.ucar.edu/cdaac/index.html), and included 2007–2011 atmprf, echprf, and sonprf products. The atmprf data is a product provided by COSMIC. It assumes that the atmosphere is dry air and provides dry temperature values from the ground to 0.2 hPa. The echprf data is obtained by interpolating the data on the ECMWF high-resolution grid onto the occultation profile, including the pressure profile, temperature profile, water vapor pressure profile, refractivity profile, and bending angle profile. The sonprf data is obtained by interpolating National Center for Atmospheric Research (NCAR)’s radiosonde data onto the occultation profile, and includes temperature, pressure, humidity and refractivity profile data. The global distribution of GPS-RO profiles is shown in Figure 1. Note that a low number of GPS-RO profiles are available in the tropics and high polar regions whereas a high number appear in the mid latitudes.

## 3. Methods

Significant changes in temperature and water vapor usually occur near the top of the MBLH, where the greatest decrease is also found in the bending angle and refractivity vertical profiles (Figure 2). However, inevitably there are high-frequency components in the angle data. If the gradient method of the bending angle is calculated using the difference method, the noise from high-frequency components will interfere with the result. In order to filter this high-frequency noise, a new numerical differentiation method is introduced below.

Suppose that the bending angle profile is a continuously differentiable function, BA(z), z is the height above sea level $z_{i}=z_{1}+(i-1)h,i=1,\cdots ,m$, where $z_{b}=z_{1}<z_{2}<\;\cdots <z_{m-1}<z_{m}=z_{t}$ is a one-dimensional uniform grid with a fixed vertical interval h. BA(z) is given by the COSMIC data angle profile, and the approximate value of the first derivative of BA(z) is obtained by the following numerical differentiation method.

According to the Newton–Leibniz formula and Simpson formula [22]:(3)$$BA(z_{i+1})-BA(z_{i-1})=\int_{z_{i-1}}^{z_{i+1}}\varphi (z)dz\approx \frac{h}{3}(\varphi_{i-1}+4\varphi_{i}+\varphi_{i+1}),\;i=2,\cdots ,m-1,$$

The matrix expression of Equation (4) is:(4)AX=b,
where,
$$A=\left(\begin{array}{ccccc} 1 & 4 & 1 & & \\ & \ddots & \ddots & \ddots & \\ & & 1 & 4 & 1 \end{array}\right),X={\left(\varphi_{1},\varphi_{2},\cdots ,\varphi_{m-1},\varphi_{m}\right)}^{T},$$
$$b={\left(\frac{3}{h}(BA_{3}-BA_{1}),\;\frac{3}{h}(BA_{4}-BA_{2}),\;\cdots ,\;\frac{3}{h}(BA_{m-1}-BA_{m-3}),\;\frac{3}{h}(BA_{m}-BA_{m-2})\right)}^{T},$$

However, if the observation data in Equation (3) has high-frequency components, the vertical gradient of bending angle will have large errors. Therefore, in order to overcome the ill-posedness of Equation (4), we transformed the solution into a problem that requires the objective functional minimum value to be solved as follows:(5)$$J(x)=\frac{1}{2}{\left\|Ax-b\right\|}^{2}+\frac{\alpha }{2}{\left\|Lx\right\|}^{2}+\frac{\beta }{2}{\left\|x\right\|}^{2},$$
(6)$$X=\underset{x}{\arg\min}J(x)={\left(\alpha L^{T}L+A^{T}A+\beta I\right)}^{-1}A^{T}b,$$
where, X is a solution that satisfies the functional minimum, α,β is the biregularization parameter that needs to be determined and the specific method is discussed below, and the matrix L is the first derivative operator, which means constraining x; thus, the last two regulation terms in Equation (5) will make the gradient oscillation of x more moderate and produce a smooth effect.
$$L={\left[\begin{array}{cccccc} -1 & 0 & 1 & & & \\ & -1 & 0 & 1 & O & \\ & O & \ddots & \ddots & \ddots & \\ & & & -1 & 0 & 1 \end{array}\right]}_{(n-2)\times n}$$

In this paper, a double-parameter model function method [23,24] was used to determine the optimal solution of the two parameters α,β in the objective functional, and then obtain the bending angle gradient. The basic technical route is as follows:

The objective function F(α,β) is:(7)$$F(\alpha ,\beta ):=\underset{x\in X}{\min}J(\alpha ,\beta ;x)=\frac{1}{2}{\left\|Ax-b\right\|}^{2}+\frac{\alpha }{2}{\left\|Lx\right\|}^{2}+\frac{\beta }{2}{\left\|x\right\|}^{2},$$

The damped Morozov deviation equation is:(8)$$G(\alpha ,\beta ):=F(\alpha ,\beta )+(\alpha^{\gamma }-\alpha )\frac{\partial F(\alpha ,\beta )}{\partial \alpha }+(\beta^{\mu }-\beta )\frac{\partial F(\alpha ,\beta )}{\partial \beta }-\frac{1}{2}\delta^{2}=0,$$
where, γ>1,μ>1 are damped coefficient and δ measures the error. Because Equation (8) is a nonlinear equation concerning α,β, solving it by the usual iterative method (such as the Newton method and quasi-Newton method) is not ideal, as they only have the property of local convergence, and the requirements for the initial value are relatively high. Therefore, we employed a model function method, which has been widely used in recent years to determine the regularization parameters. The model function method is advantageous in that the amount of calculation is greatly reduced and the convergence is guaranteed. While there are various model function options, for simplicity, we opted to use a linear model function $m_{k}(\alpha ,\beta )$ to approximate F(α,β) after k iterations, where:(9)$$\begin{array}{l} m_{k}(\alpha ,\beta ):=T_{k}+\alpha C_{k}+\beta D_{k}\to F(\alpha ,\beta ),\\ T_{k}=\frac{1}{2}{\left\|Ax_{k}-b\right\|}^{2},C_{k}=\frac{1}{2}{\left\|Lx_{k}\right\|}^{2},D_{k}=\frac{1}{2}{\left\|x_{k}\right\|}^{2}, \end{array}$$

Substituting Equation (9) into Equation (8):(10)$$G(\alpha ,\beta ):=m_{k}(\alpha ,\beta )+(\alpha^{\gamma }-\alpha )\frac{\partial m_{k}(\alpha ,\beta )}{\partial \alpha }+(\beta^{\mu }-\beta )\frac{\partial m_{k}(\alpha ,\beta )}{\partial \beta }-\frac{1}{2}\delta^{2}=0,$$

For the sake of simplicity, let γ=2,μ=2, the solution of Equation (10) is obtained as follows:(11)$$\begin{array}{l} a_{k+1}=\sqrt{\frac{\left\|\frac{\delta^{2}}{2}-T_{k}-D_{k}\beta_{k}\right\|}{C_{k}}}\\ \beta_{k+1}=\sqrt{\frac{\left\|\frac{\delta^{2}}{2}-T_{k}-C_{k}\alpha_{k+1}\right\|}{D_{k}}} \end{array}$$

Let $\varepsilon =10^{-4}$, if the following conditions in Equation (12) are not met, bring $\alpha_{k+1},\beta_{k+1}$ back to Equation (10), and then use Equation (11) to calculate $\alpha_{k+2},\beta_{k+2}$. Repeat this cycle until Equation (12) is satisfied.
(12)$$\begin{array}{l} \frac{\left|\alpha_{k+1}-\alpha_{k}\right|}{\alpha_{k+1}}<\varepsilon \\ \frac{\left|\beta_{k+1}-\beta_{k}\right|}{\beta_{k+1}}<\varepsilon \end{array}$$

By putting the $\alpha_{k},\beta_{k}$ that meet the conditions in Equation (12) into Equation (6), the gradient of bending angle X is obtained. After that, the MBLH can be obtained by the maximum gradient method. Compared with the central difference method, this method requires more calculations, but it has a filtering effect on the high-frequency interference in the bending angle profile, which makes the obtained MBLH more accurate.

## 4. Climate Characteristics of MBLH

### 4.1. Annual Variation Characteristics

Figure 3 shows the characteristics of the distribution of the average MBLH in the global ocean. Figure 4 shows the global distribution of sea breezes and sea surface temperature (SST). We define $Cu=-\nabla \vec{T}/v$ as a quantitative representation of ocean currents. When $-\nabla \vec{T}/v>0$ it represents warm currents and $-\nabla \vec{T}/v<0$ it represents cold currents, wherein $\nabla \vec{T}=\frac{\Delta \vec{T}}{2}$ represents the variations in the SST variations along the eV direction. Figure 5 shows the correlation coefficients between the MBLH anomaly relative to the average MBLH in the range of 30° × 30° and $Cu=-\nabla \vec{T}/v$ within four regions (A, B, C, D). The correlation coefficients of the four regions are 0.87, 0.58, 0.83 and 0.71, respectively. We can conclude that the MBLH over the ocean correlates closely with cold or warm ocean currents. On the west coast of the mainland, cold ocean currents overlap with areas of low boundary layer height. These areas are accompanied by subsidence airflow, and the height of the inversion layer is relatively low; thus the MBLH obtained from the minimum angle gradient is low. For the same reason, on the east coast of the mainland, the warm ocean currents overlap with the high-value area at the boundary layer. The MBLH in the low latitude region is about 2.5–3 km, and the MBLH in the high latitude region is about 1 km. Essentially, there are more solar radiation hours in the low latitude region, and the rising air mass carries a large amount of condensation latent heat, which can reach the height of the troposphere. In contrast, the underlying surface in the high latitude region is stable and cold, and the corresponding boundary layer height is low. Thus, the MBLH gradually decreases from the equator to the poles.

### 4.2. Seasonal Variation Characteristics

The numerical regulation method, in combination with the atmprf and echprf products from CDAAC, were used to comprehensively analyze the seasonal variation characteristics of the height of the global ocean boundary layer. Figure 6 shows the monthly average boundary layer height (h_atm) obtained from atmprf data. Note that the ocean areas near the Brazilian, California, Canary, and Bengra cold currents have a lower MBLH. This can be explained by the fact that the underlying surface temperature is lower than that of the nearby sea area and the sea surface wind blows from a low temperature to a high temperature along the direction of the temperature gradient, as depicited in Figure 7, from which we can also see that the MBLH is proportional to the wind speed from high to low SST and the SST gradient. For the northern and southern equatorial currents, the boundary layer is highest in April. In the regions with a westerly current in the southern hemisphere, the MBLH is stable and stays at ~2 km. In contrast, in the westerly current region of the northern hemisphere, the westerly current is blocked to the west due to the existence of the continent, and therefore forms a cold and warm current. The MBLH on the west coast of the continent is lower at ~1 km, and on the east coast of the continent, it is higher at ~2 km. In January and April, the MBLH is basically situated below 1 km in the Arctic, and at 1–1.5 km in the waters around Antarctica, while in July and October, the MBLH is located at 1.5–2 km in the Arctic, and at ~1 km in the waters near Antarctica.

Figure 8 shows the monthly average boundary layer height (h_ech) obtained from echprf data. Note that the MBLH reaches a height of 3 km in January and April, especially in the area of westerly ocean current in the southern hemisphere. On the contrary, the MBLH is lower in July and October than the heights observed in January and April. With the exception of the region of westerly ocean currents where the MBLH can reach 2.5 km during the months of July and October, the MBLH remains below 2 km in all the remaining regions. The explanation for this phenomon is as follows. In winter, the ocean surface is warm, and thus the water vapor rises and releases latent heat at a certain height. This results in the formation of the strongest inversion layer produced throughout the four seasons [17], which pushes the MBLH higher. In contrast, during summer, the ocean surface temperature is lower. Thus, the heating mass in the ocean moves to the cold ocean surface, and easily forms a low-temperature inversion layer, thereby maintaining a relatively low MBLH. The MBLH gradually increases across the mainland from the low stratus area on east coast to the deep convective area on the west coast—especially in the ITCZ area where the MBLH can reach 2.5 km during January and April. These results are depicted in Figure 8, which shows that the west coast of the continent, where the cold current passes by (e.g., Brazil cold current) corresponds to the area with the lower MBLH, and the area corresponds to the direction of the cold current flow. The direction of the current up the coast correlates with the wind direction. For the westerly jet, the North American mainland coastline and the European coast determines whether it is the California cold current or the North Atlantic warm current.

Figure 9 shows the gap between h_atm and h_ech. Note that in January and April, h_atm is higher in the low latitude areas, such as the ITCZ area, while h_ech is higher in the southern hemisphere’s westerly jet region and in the northern hemisphere’s middle and high latitudes. In July and October, the convection activities in the northern hemisphere are strong. Thus, h_atm is higher than h_ech, and the gap between the two in the southern hemisphere decreases. In summary, these results imply that h_atm is more sensitive to convective cloud top capture, and is generally higher than h_ech where convection is active.

Figure 10 shows the distribution of the standard deviation of the atmprf data. Note that the standard deviation value is very high (>0.5) in the low-latitude oceans, especially in the ITCZ area and regions dominated by and downwind from the subtropical trade winds. In contrast, the January and April mid-high latitude standard deviations are low, while the July and October high and low latitude standard deviations are high. A possible explanation is that convection is active in low-latitude areas, thus, there are multiple inversion layers. High-latitude regions are mostly high-level clouds that are uniform and thin. At the same time, comparing Figure 1 and Figure 10 shows that the number of occultation profiles also greatly affects the standard deviation of the results.

### 4.3. Interannual Variation Trend in MBLH and a Case Study in the South China Sea 

Two primary modes and the time coefficients of global MBLH were analyzed by Empirical Orthogonal Function (EOF). As shown in Figure 11a, the variance contribution rate of the first mode reached 96%. The MBLH decreased from low latitude to high latitude, from the west coast of the mainland to the westward, and corresponded to cold and warm ocean currents, respectively. The time series of the first mode is seasonally distributed. As shown in Figure 11b, the variance contribution rate of the second mode reached 2.2%. Note that the MBLH anomaly spatial distribution is opposite in the northern and southern hemispheres. The time coefficient demonstrates that it is related to the north-south movement of the direct sun point.

Figure 12 shows the MBLH interannual variability from 2007–2011 in parts of the South China Sea (latitude range: 15° N–25° N, longitude range: 105° E–115° E). The numbers were derived from 2007–2011 COSMIC atmprf, echprf, and sonprf data. The bending angle served as the variable in the first two sources, and refractivity in the latter. The results show that the MBLH has undergone a downward trend over the past 5 years, and parallel those reported by Guo, et al. [5] for the BLH trend over mainland China. The MBLH obtained by the atmprf data is the largest, the result obtained by the sonprf data is the smallest; and the atmprf results approximate those obtained using echprf. This figure also shows that the MBLH over the South China Sea is highest in January and lowest in July. In January, the South China Sea wis in the ascending branch of the land-sea circulation when convection is active, which is conducive to development of the boundary layer height. In July, it is in the sinking branch, and the lower temperature inversion layer is not conducive to boundary layer development.

## 5. Conclusions

We estimated the global MBLH using GPS-RO profiles from the FORMOSAT-3/COSMIC satellites using a new numerical differential regularization method to analyze and compare the MBLH based on three CDAAC products—atmprf, echprf, and sonprf. Compared with the central difference method, this new method can better filter high-frequency noise to obtain more accurate results, but with high computation cost. The annual variation characteristics of MBLH were studied using this method, and we found that the average MBLH is closely correlated with cold and warm ocean currents. The MBLH is relatively low in the sea area near cold ocean currents, and relatively high in the sea area near warm ocean currents. We also investigated and compared the seasonal variation characteristics of MBLH obtained by atmprf and sonprf. The results show that the seasonal variation in the MBLH is related to the north-south movement of solar direct points and the distribution of ocean currents, which was also verified through EOF decomposition of the global MBLH. The comparison between atmprf and echprf demonstrates that atmprf is more sensitive to convective cloud top capture. Finally, the interannual changes in parts of the South China Sea indicate that the MBLH exhibited a downward trend from 2007 to 2011.

The numerical regularization method proposed in this work is suitable for fitting functions of discrete data, and can effectively avoid data errors, thus providing stable and reliable results. With further work on the modification of this method, we expect the MBLH found by the method proposed in this paper could be incorporated into climate models to significantly improve the accuracy of future climate forecasting.

## Figures and Tables

**Figure 1 sensors-20-04762-f001:**
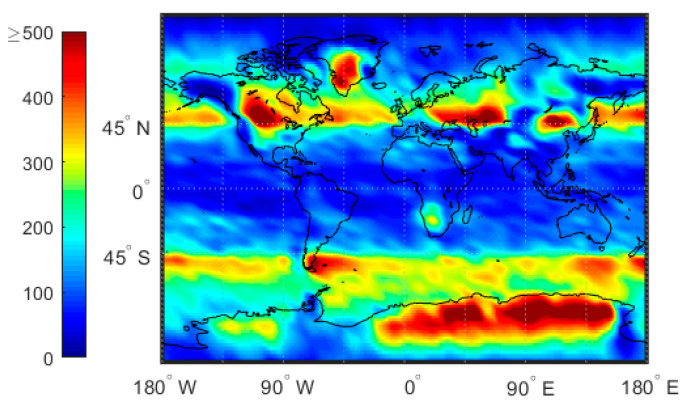
The number of COSMIC GPS-RO profiles from 2007 to 2011 for which the signal penetrates to below 500 m above the surface. The color bar represents the number of profiles.

**Figure 2 sensors-20-04762-f002:**
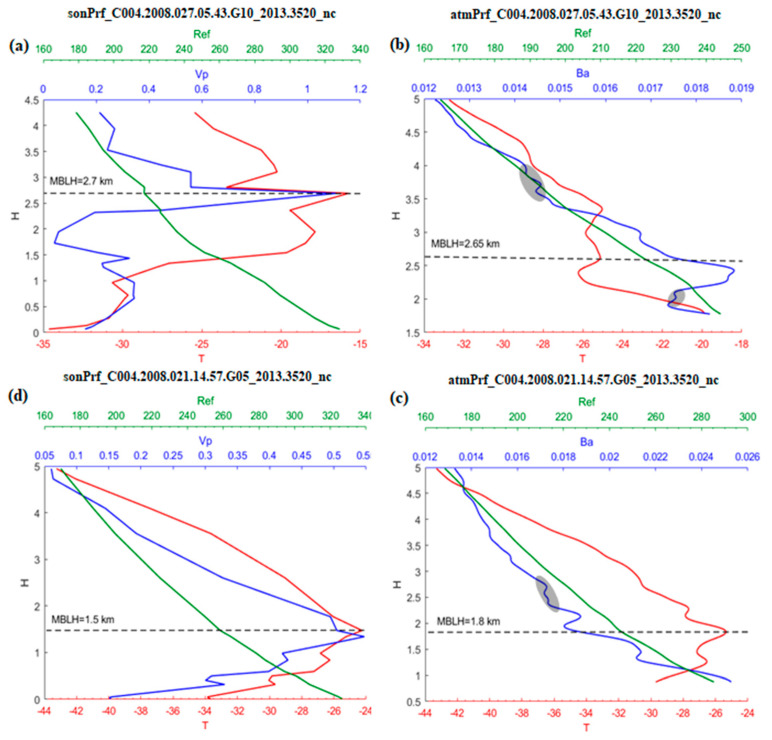
(**a**,**c**) are sonprfs, (**b**,**d**) are atmprfs. (**a**,**b**) matched at (67.3° N, 87.9° W); (**b**,**d**) matched at (79.2° N, 59.3° W). Ref, VP, T, Ba denotes refraction, vapor, temperature, bending angle, respectively. The dashed line represents the height of the inversion layer in the temperature profile. The elliptical shaded region represents the high frequency part of the occultation bending angle profile.

**Figure 3 sensors-20-04762-f003:**
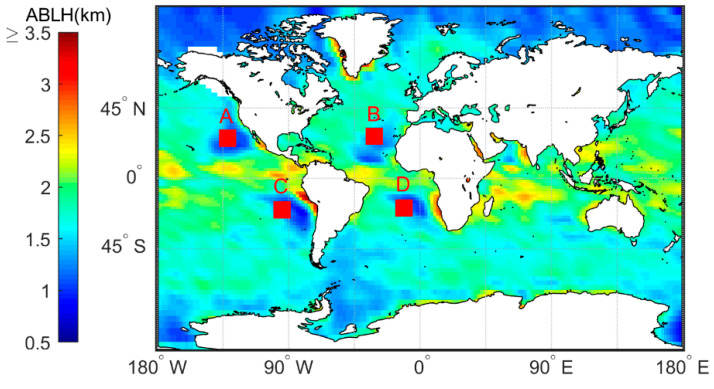
Annual marine boundary layer height (MBLH) based on the 2007–2011 CDAAC “atmprf” bending angle products. The small red boxes (A, B, C, D) in the figure have a range of 10°×10°.

**Figure 4 sensors-20-04762-f004:**
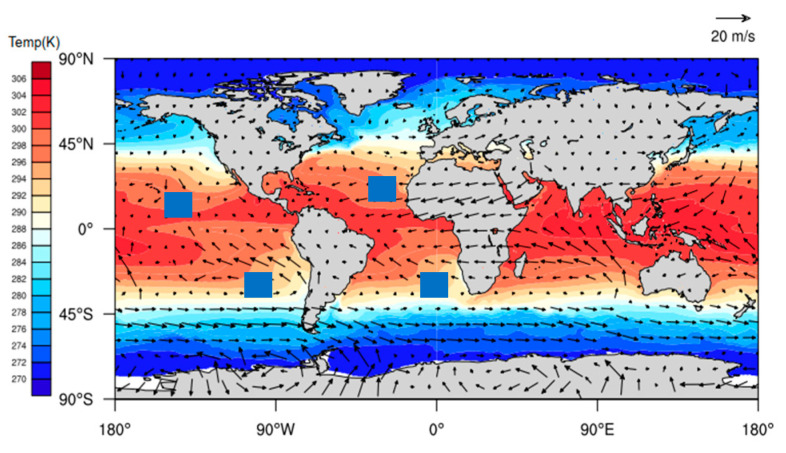
The annual mean wind-driven current represented by the annual mean sea surface temperature field and wind field. The black boxes indicate cold currents. The small blue box. corresponds to the four A, B, C, D areas in Figure 4.

**Figure 5 sensors-20-04762-f005:**
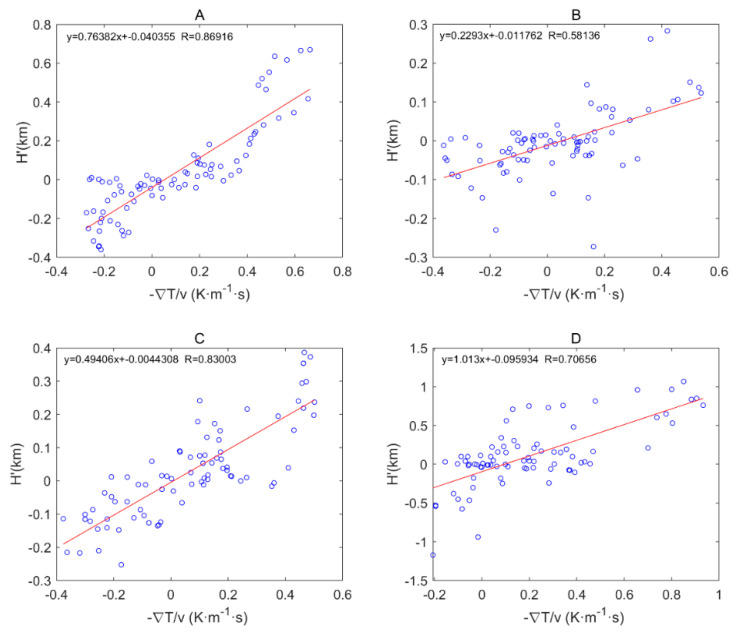
Correlation coefficients between MBLH anomalies relative to the average MBLH in the range of $30^{{}^{\circ}}\times 30^{{}^{\circ}}$ and $Cu=-\nabla \vec{T}/v$ in the four regions of (**A**–**D**).

**Figure 6 sensors-20-04762-f006:**
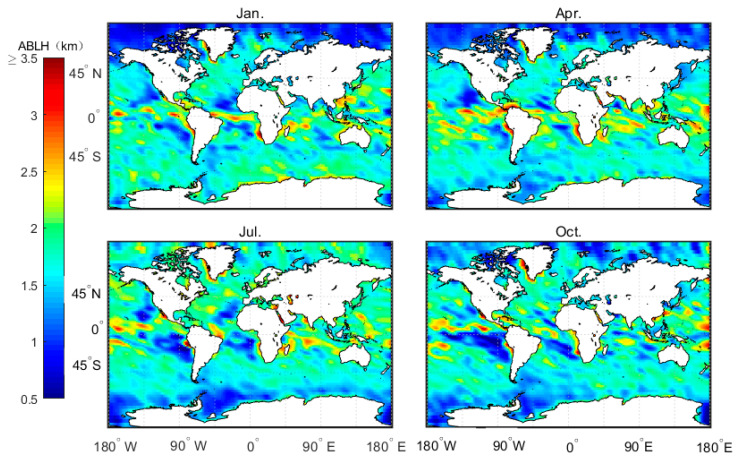
Average MBLH over four months based on the 2007–2011 CDAAC atmprf bending angle products.

**Figure 7 sensors-20-04762-f007:**
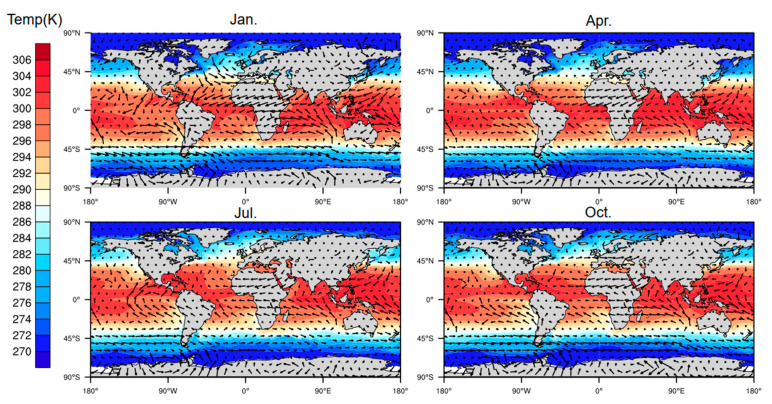
The perennial average monthly wind-driven current represented by the monthly mean sea surface temperature field and wind field.

**Figure 8 sensors-20-04762-f008:**
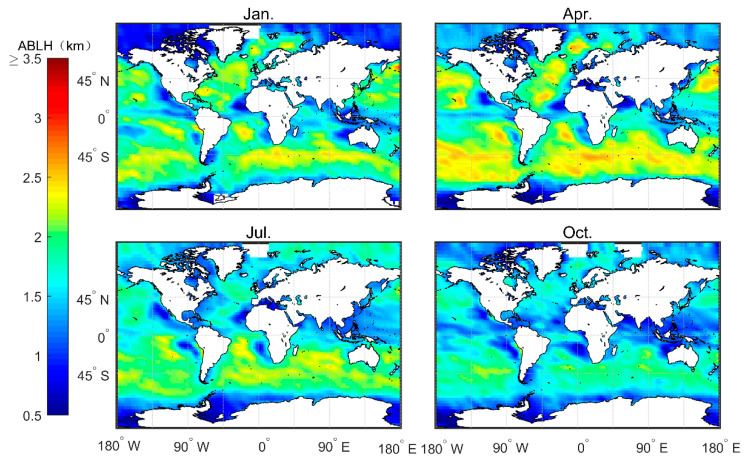
Average MBLH over four months based on the 2007–2011 CDAAC “echprf” bending angle products.

**Figure 9 sensors-20-04762-f009:**
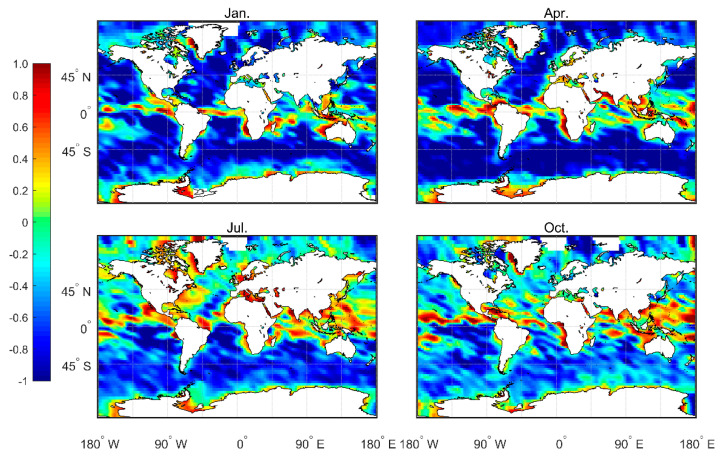
The difference between average MBLH based on 2007–2011 CDAAC echprf and atmprf products.

**Figure 10 sensors-20-04762-f010:**
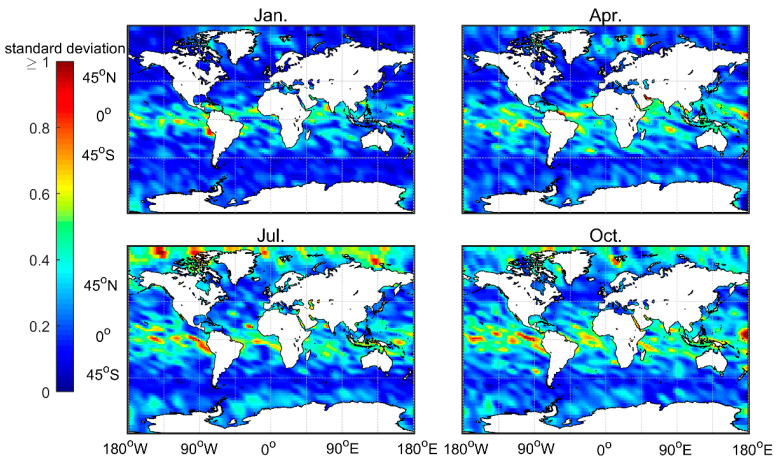
The distribution of the standard deviation for the average MBLH over four months, based on COSMIC’s 2007–2011 atmprf bending angle data.

**Figure 11 sensors-20-04762-f011:**
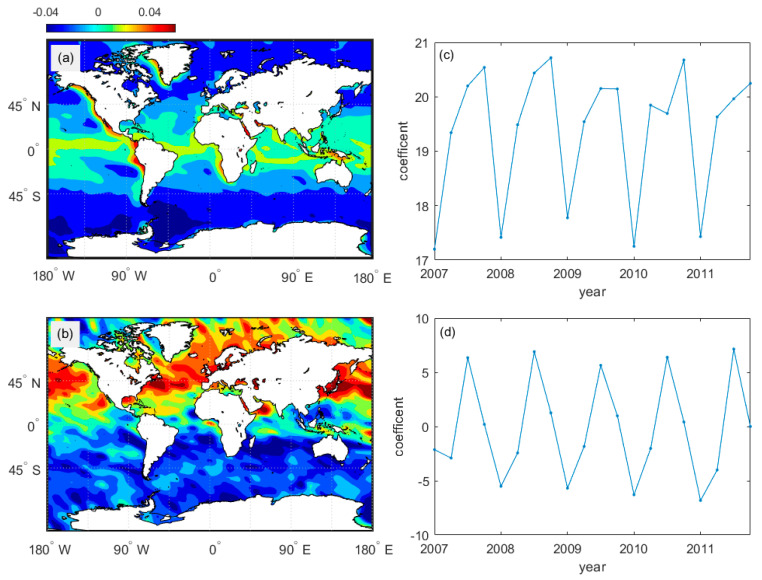
(**a**,**b**) The two modes of MBLH global distribution from 2007 to 2011. The colorimetric region is the MBLH anomaly; (**c**,**d**) The time coefficient corresponding to two modes.

**Figure 12 sensors-20-04762-f012:**
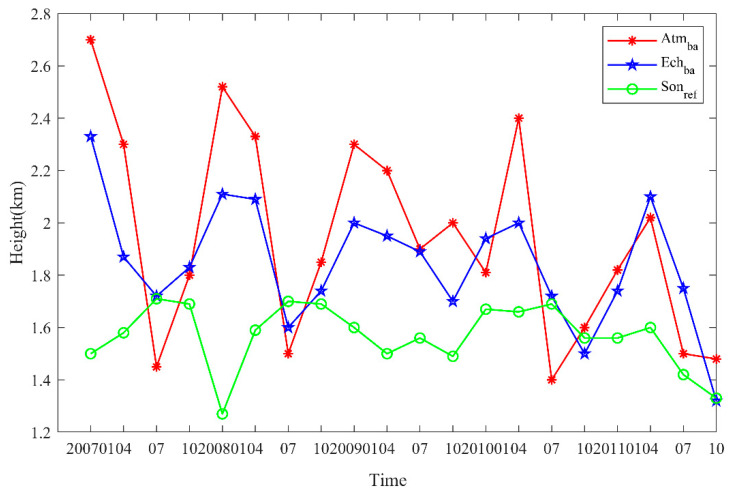
The interannual variability of MBLH in the waters near the South China Sea was derived from COSMIC’s atmprf and echprf bending angle and sonprf refractivity.

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
