# Peer review of "Marine Boundary Layer Height Obtained by New Numerical Regularization Method Based on GPS Radio Occultation Data"

_sensors, 2020, doi:10.3390/s20174762_

Round 1

Reviewer 1 Report

It is a pity that I donot think the manucript can be published in its present form. More work are needed to improve.

  1. Line 20-21, what is the Quantitative analysis of correlation?
  2. The abstract part should be revised. In its present form, it is not satisfied the requirement of scientific paper.
  3. line 41-44, what is the merits and drawback of the different methods?
  4. Line 88-101, for this part, it can be included in the introduction part. this section should pay attention to data description
  5. the title for section 2 and section 3 is data and methods, respectively. However, it is too fuzzy to summarize your work. please check and revise.
  6. Equation3, please give the reference
  7. Figure 3, I donot think that can using these equations to represent the flowchart is appropriate.
  8. Title for 4.1 is subsection, please revise this part.
  9. The conclusion part should be revised. Conclusion part should summarize your work .

Reviewer 2 Report

The paper "Marine Boundary Layer Height obtained by New Numerical Regularization Method Based on GPS Radio Occultation Data" is written quite well. The paper describes about methods and results nicely. My few concerns are as below:

1. I suggest authors to highlight why this new method is better than the existing ones? What are the advantages of using the proposed method than those already in literature. Also please mention any short-comings of the proposed method.

2. In the paper, authors have clearly mentioned about different literature that calculated the boundary layer height for different regions (tropical, European region) [ref to 53-73 of the manuscript]. I strongly suggest authors to include a section where they compare the result from the proposed method and the result from the literature. Authors can then explain what are the contributions of this new method and its applicability.

3. In Figure 1 caption, authors mention "The number of COSMIC GPS-RO profiles from 2007 to 2011 for which the signal 117 penetrates to below 500 m above the surface". Here the color bar represents height or number?

4. For Figure 2, please explain all the short forms used; T, Ba, Vp, Ref. And in the figure caption it is mentioned "The elliptical shaded region represents the high frequency part of the occultation bending angle profile". Can authors point this out clearly.

5. Section 4.1, it is written as "subsection". Please update accordingly.

6. Section 4.3, line 282, there is typo "two2".

Round 2

Reviewer 1 Report

The authors have responded my comments quite well.

I have no any other comments now.